# Switching the activity of Cas12a using guide RNA strand displacement circuits

Lukas Oesinghaus[1] & Friedrich C. Simmel [1]

The CRISPR effector protein Cas12a has been used for a wide variety of applications such as in vivo gene editing and regulation or in vitro DNA sensing. Here, we add programmability to Cas12a-based DNA processing by combining it with strand displacement-based reaction circuits. We first establish a viable strategy for augmenting Cas12a guide RNAs (gRNAs) at their 5′ end and then use such 5′ extensions to construct strand displacement gRNAs (SD gRNAs) that can be activated by single-stranded RNA trigger molecules. These SD gRNAs are further engineered to exhibit a digital and orthogonal response to different trigger RNA inputs—including full length mRNAs—and to function as multi-input logic gates. We also demonstrate that SD gRNAs can be designed to work inside bacterial cells. Using such in vivo SD gRNAs and a DNase inactive version of Cas12a (dCas12a), we demonstrate logic gated transcriptional control of gene expression in *E. coli*.

[1] Physics Department E14, Technical University Munich, 85748 Garching, Germany. Correspondence and requests for materials should be addressed to F.C.S. (email: simmel@tum.de)

Toehold-mediated strand displacement (SD) reactions are the most widely used dynamic processes in nucleic acid nanotechnology and molecular programming[1]. They have been utilized for the operation of nucleic acid based nanodevices[2], dynamical systems[3–5], for the realization of DNA molecular computers[6,7] and robots[8], and also as the basis of a wide variety of biomolecular sensors[9]. Strand displacement reactions involve the invasion of a DNA or RNA duplex structure by a third strand which is—at least partly—complementary to one of the two duplex strands. Repeated hybridization and strand displacement reactions can be used to reversibly switch nucleic acid structures between alternative conformations. The presence of a single-stranded toehold sequence adjacent to the target duplex, at which the strand displacement reaction can be initiated, considerably speeds up the process[10]. Among the main advantages of the strand displacement technique over other biomolecular switching processes is its sequence-programmability that is derived from the predictable nature of Watson-Crick base-pairing interactions, which also allows for the implementation of in silico structure prediction and interaction design tools[11].

Over the past few years, strand displacement techniques have been increasingly utilized for the control of biologically relevant processes, both in vitro as well as in vivo. In this context, naturally occurring, sequence-based RNA regulatory mechanisms, such as transcriptional and translational riboregulators, RNA interference with small interfering RNAs, or CRISPR are particularly promising[12–15]. DNA strand displacement reactions involving chemically stabilized DNA were used for in vivo computation coupled to gene knockdown by RNA interference in mammalian cells[16,17]. RNA strand displacement has been used to enhance and extend the functionality of engineered RNA regulators, as demonstrated with the construction of de novo-designed riboregulators called toehold switches. In earlier riboregulator systems, the activity of transducer strands responsible for transcription or translation was controlled by loop–loop or loop–linear interactions with cognate trans-acting RNAs. By replacing this interaction with a linear–linear toehold-mediated strand displacement reaction, the dynamic range and orthogonality were improved significantly[18]. Furthermore, the programmability of strand displacement allows for the implementation of RNA sensors and complex logic functions[18,19]. There are, however, limitations to using this type of translational control. To create layered logic gates, the input of a gate needs to have the same form as the output, i.e., an RNA input needs to produce an RNA output. Furthermore, translational control requires engineering of the 5′ untranslated region of the mRNAs and is thus difficult to apply to naturally occurring gene products.

In this respect, CRISPR-mechanisms based on CRISPR-associated (Cas) proteins and guide RNAs (gRNAs) offer a powerful alternative for RNA-based gene regulation. In the most common implementation, a catalytically inactive version of Cas9 nuclease (dCas9) is targeted at a gene by setting the variable protospacer domain of its gRNA to be sequence complementary to the intended binding site on the DNA target, thereby repressing transcription[14]. By fusing dCas9 with transcription activators or repressors, this general mechanism has also been adapted to transcription regulation in eukaryotes[20,21]. Since two gRNAs in combination with dCas9 function as a NOR gate when aimed at the same target, layered logic circuits can be constructed from sets of Cas9 gRNAs[15]. Furthermore, other CRISPR-associated proteins, such as Cas12a (formerly Cpf1) and Cas13a, have been used to construct in vitro detection systems for both DNA and RNA targets[22–24].

Here, we combine the programmability of strand displacement with the capabilities of the CRISPR nuclease *Acidaminococcus* Cas12a (AsCas12a) by constructing AsCas12a gRNAs that are switchable via RNA strand displacement (which we termed strand displacement gRNAs or SD gRNAs). Cas12a has a variety of features that differentiate it from the more commonly used Cas9: Its gRNAs are comparatively short, it has a T-rich proto-spacer adjacent motif (PAM), and it creates staggered rather than blunt ends when cutting a target DNA[25]. Next to its nuclease activity toward the target dsDNA, it displays two additional enzymatic functions: (i) it processes its own gRNAs and (ii) shows an unspecific ssDNase activity upon target binding that can be used for sensitive enzymatic detection of DNA targets[23,26–28]. In the following, we will make use of all three of these enzyme functions.

We begin by investigating strategies for the extension of Cas12a gRNAs and find that only extensions at the 5′ end reliably result in high activities. We then demonstrate the concept of SD gRNAs, in which binding of Cas12a is suppressed by occluding the handle domain of the gRNA via hybridization with a partially complementary extension at the 5′ end. Upon addition of a complementary RNA trigger molecule, the occluding domain is unfolded via toehold-mediated strand displacement, which facilitates Cas12a binding and thus processing of the gRNA. The ensuing cleavage of the SD gRNA upstream of the handle region results in the recovery of a fully active, regular-length Cas12a gRNA. We next introduce orthogonality between different gRNA triggers, which is achieved by covering also part of the gRNA target sequence by the occluding domain. The resulting orthogonal SD gRNAs show close to digital response to their respective trigger RNAs–in the absence of the cognate trigger RNA, only minimal activity is observed, while in its presence the target DNA is fully processed.

We then proceed to demonstrate the potential of SD gRNAs for applications in molecular computing, sensing, and control of bacterial gene expression. Multi-input SD gRNAs are created by splitting the trigger RNA into multiple parts. This is further used to implement Cas12a-based logic circuits, which we demonstrate with the realization of orthogonal two-input AND gates and a three-input AND gate. We also use the basic architecture of a two-input AND gate to demonstrate how natural RNA sequences —such as mRNA—can be used as inputs for SD gRNAs. As an in vivo application, we demonstrate that SD gRNAs can regulate bacterial gene expression in E. coli using a DNase-dead mutant of Cas12a (dCas12a). Upon activation, the repression efficiency of SD gRNAs is as high as those of regular-length gRNAs, leading to dynamic ranges of up to 100 for efficient gRNA target sequences. This also allows the implementation of SD gRNA-based NAND gates for the logical control of gene expression in E. coli.

## Results

**Extension strategies for Cas12a gRNAs**. With a length of ≈40 nt, AsCas12a gRNAs are relatively short and have a comparatively simple secondary structure consisting of a short handle domain containing a single stem loop followed by the target domain[25]. Hence, there are three basic possibilities for an extension of the gRNA sequence, namely at the 5′ end, within the stem loop and at the 3′ end (Fig. 1a). A special feature of a 5′ extension is the recovery of regular-length gRNA upon binding due to Cas12a's inherent RNA-processing activity[26].

In order to test the three extension possibilities, we assayed the nuclease activity of the extended gRNAs via their ability to cut a fluorescently labeled target (Supplementary Fig. 1). We designed sequences for purely single-stranded and partially double-stranded 5′ and 3′ extensions as well as double-stranded extensions at the stem. The expected folding of the designed sequences was predicted using NUPACK[29]. Extending the stem abolished the cutting activity for both tested variants (Fig. 1b), a

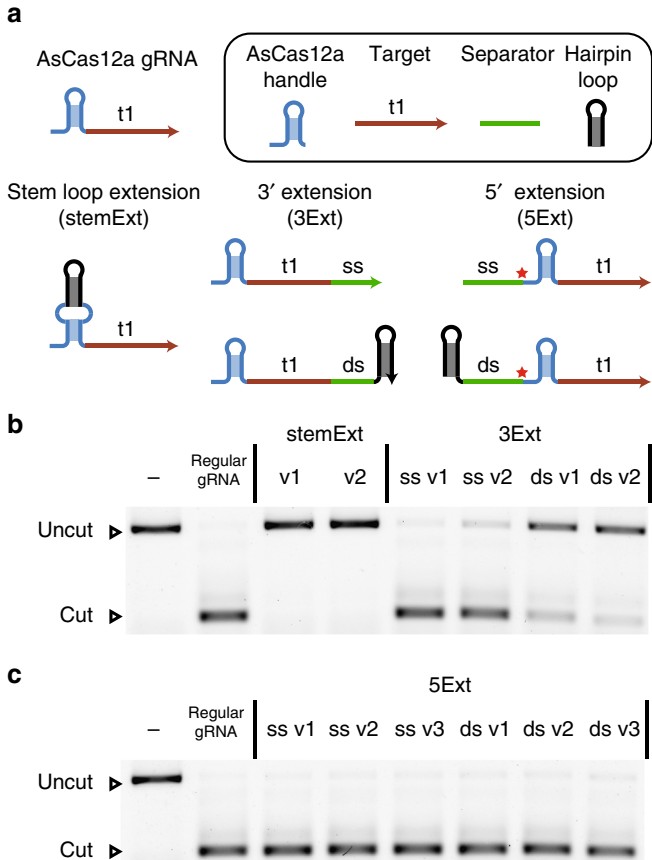

**Fig. 1** Possibilities for extending Cas12a gRNAs. **a** The principal possibilities are extension of the stem (stemExt), extension at the 3′ end (3Ext), and extension at the 5′ end (5Ext). The extension of the stem is necessarily double stranded, while at the 3′ and 5′ end either a single-stranded (ss) extension or a (partially) double-stranded (ds) extension can be added. The red star for the 5′ extension marks the point where Cas12a cleaves a successfully bound gRNA. **b** Agarose gel of Cas12a cutting a fluorescently labeled target DNA for different stemExt and 3Ext gRNAs with target sequence t1 (v1: version 1, uncut: 1190 bp, cut: 357 bp). **c** Agarose gel of Cas12a cutting a fluorescently labeled target DNA for different 5Ext gRNAs with target sequence t1 (uncut: 1190 bp, cut: 357 bp). A detailed description of the gRNA designs can be found in Supplementary Notes 1 and 2 and Supplementary Fig. 10 and 11. Source data are provided as a Source Data file

result consistent with the findings of Li et al.[30]. Extending the gRNA at the 3′ end resulted in a modest reduction of activity for an exclusively single stranded extension. However, addition of a hairpin resulted in a dramatic reduction in activity, even when the hairpin structure was placed 10 nt outside of the assumed Cas12a binding region to avoid a potential steric hindrance (Fig. 1b)[31].

In contrast, extensions at the 5′-end retained comparatively strong Cas12a activity for all tested designs. For single-stranded extensions, we tested a random, unstructured single-stranded sequence (ss v1), as well as sequences corresponding to the full repeat of AsCas12a (ss v2) and the full repeat of *Francisella novicida* Cas12a (FnCas12a) (ss v3)[25]. As extensions containing double-stranded motifs, we tested a hairpin directly preceding the handle sequence (ds v1), a hairpin separated from the handle by a 3 nt single-stranded domain (ds v2), and a large hairpin with a single-stranded domain corresponding to the full AsCas12a repeat (ds v3). In all cases, the target sequence was efficiently cut (Fig. 1c). This indicates substantial freedom in choosing the single-stranded domain for 5′ extensions. Also when changing the target sequence, the ability of 5′ extended gRNAs to promote

cutting of the target by Cas12a was retained (Supplementary Fig. 2). In the following, we refer to the sequence domain between 5′ hairpin and handle sequence as the separator domain. For subsequent experiments, we used a separator sequence corresponding the full AsCas12a repeat.

**Constructing strand displacement switchable gRNAs.** Based on the successful 5′ extension design we next constructed gRNAs switchable by a strand displacement mechanism. To this end, we attached a switch domain complementary to parts of the separator and the handle to the 5′ end of the separator, resulting in a secondary structure in which the gRNA handle is disrupted (Fig. 2a) and Cas12a binding is suppressed. An additional toehold domain (X1) at the 5′ end of the modified gRNA facilitates the binding of a trigger RNA[32], which restores the secondary structure of the Cas12a handle via toehold-mediated strand invasion into the switch domain. This allows Cas12a to bind and cleave off the 5′ end, producing a regular-length, fully active gRNA. We refer to this design as the handle (hd) design, since the switch domain is simply complementary to the handle of the gRNA but does not include the target domain.

As shown in Fig. 2b, the strand displacement (SD) gRNA behaves as expected. In the absence of trigger RNA, Cas12a does not cut its target. In the presence of trigger, the target is fully cut. This works equally well for gRNAs with two different target sequences t1 and t2. As explained schematically in Fig. 2a, the switching mechanism utilizes the intrinsic RNA-processing capability of Cas12a. Upon addition of the trigger, Cas12a cleaves the SD gRNA into two fragments, one of which is of the same length as a regular gRNA, which is also observed experimentally (Fig. 2c). We tested the activity of the SD gRNA using a cutting assay. For a target concentration of 10 nM, addition of 10 nM of trigger already leads to significant (>60%) target cutting, and 15 nM of trigger results in around 90% cut target (Fig. 2d, e).

For activation assays, we used four times as much SD gRNA as target DNA to show that there is only a small leak reaction. The activation assay when using the same amount of SD gRNA as target DNA is very similar for low target concentrations, but saturates at around 90% cut target for larger amounts of trigger (Supplementary Fig. 3). In effect, one molecule of trigger approximately leads to the cutting of one target molecule, which is consistent with the previous finding that each active Cas12a molecule binds and cleaves only one target DNA[23,33].

**Design considerations for trigger RNAs and SD gRNAs.** A variety of considerations went into the specific design of the SD gRNA and trigger RNAs shown in Fig. 2, which are discussed extensively in Supplementary Note 3 and Supplementary Fig. 12. Consistent with earlier studies, we found that transcribing short, structure-free RNAs with T7 polymerase can result in incorrect products due to accidental extension of run-off transcripts[34]. All trigger RNAs were therefore equipped with hairpins at the 5′ and 3′ end (omitted in the Figures for clarity), which improved their transcription efficiency and also protected them from degradation by RNases[35]. A discussion of the challenges encountered during in vitro transcription of short RNAs and how to resolve them can be found in Supplementary Note 4 and Supplementary Fig. 4.

Quite generally, control of secondary structure is critical to the functionality of strand displacement circuits. In particular, a lack of secondary structure in the trigger and the toehold is necessary to ensure fast strand displacement kinetics[32,36]. To appropriately design the sequences of our SD gRNAs/trigger RNA we used the nucleic acid structure prediction tool NUPACK[29,37], and to independently verify the structure of the designed RNA strands

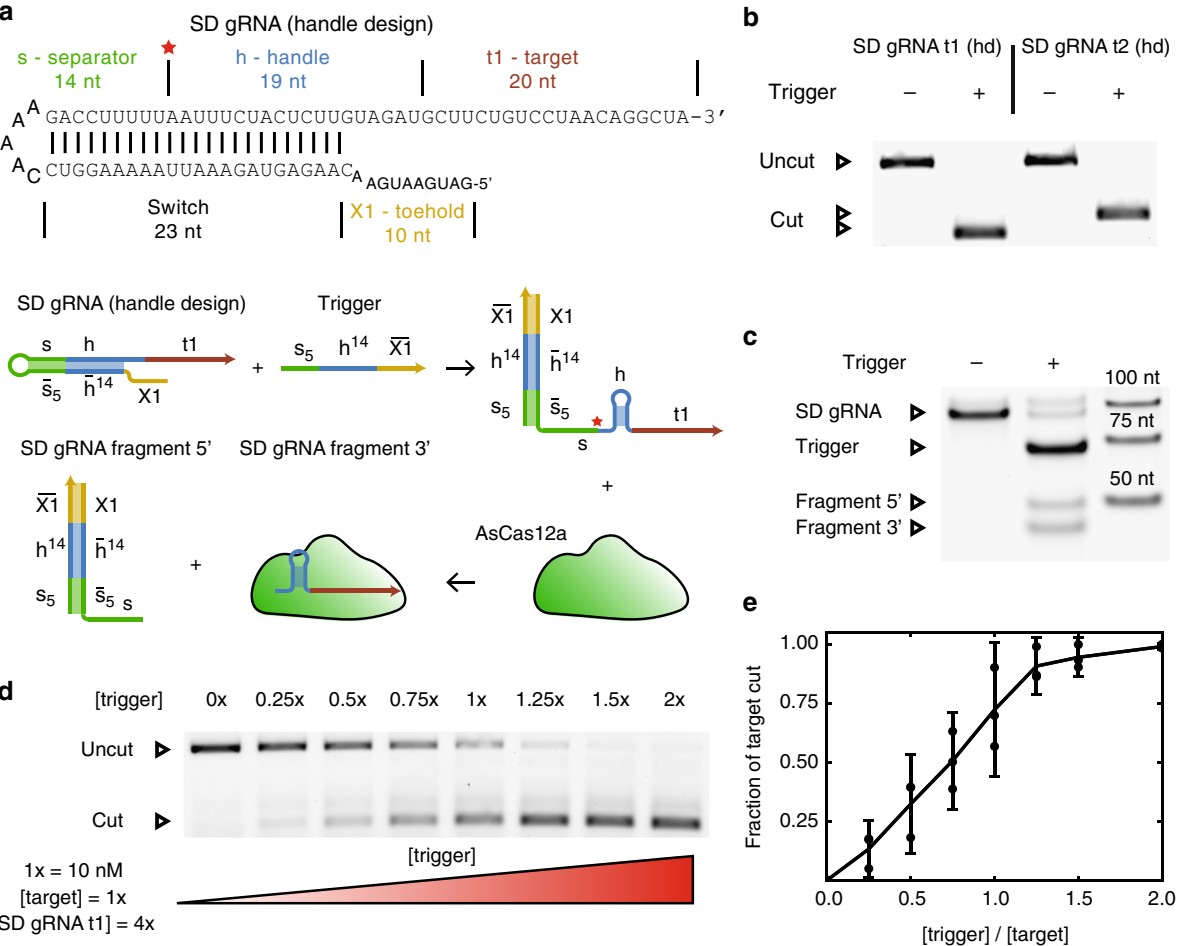

**Fig. 2** Principle of strand displacement switchable gRNAs. **a** When the RNA trigger binds the SD gRNA, the 5′ extension domain occluding handle is displaced, thereby restoring the gRNA handle. Binding of Cas12a leads to cleavage of the gRNA, which removes the 5′ extension and creates an active Cas12a-gRNA complex. The domains are labeled as follows: s—separator, h—handle, t—target, X1—toehold. The overall nomenclature follows Zhang et al.[32]: A domain is denoted by a single letter. An upper index $x$ denotes the first $x$ nucleotides of a domain counting from its 5′ end. A lower index $x$ denotes all but the first $x$ nucleotides of a domain counting from its 5′ end. A combination of upper and lower indices includes those nucleotides that are present in both subdomains. A bar above the letter marks the reverse complement of a domain. The red star marks the position where Cas12a cleaves a successfully bound gRNA. **b** Agarose gel showing cutting of a target DNA by handle-based SD gRNAs with two different target sequences in the absence and presence of trigger RNA (uncut: 1190 bp, cut t1: 357 bp, cut t2: 506 bp). **c** Denaturing PAGE showing induced gRNA processing due to trigger binding. **d** Agarose gel showing activation of target cutting by varying amounts of trigger RNA (uncut: 1190 bp, cut: 357 bp). **e** Transfer function derived from the fraction of cut target by gels as shown in (**d**) ($N = 3$, $t$-distribution two-sided 90% confidence interval). Source data are provided as a Source Data file

we used the mfold package[38]. NUPACK design files are provided as Supplementary Software.

Of note, a special role in the design of our RNA strand displacement processes was taken by the GU wobble base pairs, which introduce both constraints and opportunities to the design process. On the one hand, wobble base pairs render the trigger and toehold more likely to form unwanted secondary structure. On the other hand, explicitly allowing for wobble base pairs provides additional freedom in the design of nominally fixed sequences such as that of the gRNA handle, which can be critical, e.g., for the realization of a structure-free trigger. Since the handle contains a hairpin and the fraction of the handle sequence that is paired with the switch domain is reproduced in the trigger—up to GU pairs between the handle and the switch and between the switch and the trigger—pairing the complete handle hairpin would necessitate the use of a large number of GU pairs to obtain a completely structure-free trigger. To avoid this issue, we only paired the first 14 nt of the handle domain in the design of the SD gRNAs. The 9 nt of the separator that are paired with the switch

domain ensure that a sufficiently strong secondary structure is present on both sides of the Cas12a gRNA processing site.

**Target-based SD gRNAs for making triggers orthogonal**. For more advanced applications that involve the operation of several processes in parallel, e.g., in multi-layered reaction circuits, multiple independent and orthogonally switchable SD gRNAs are desirable. To achieve orthogonality in trigger activation, mutually non-interfering sequence domains are required. In handle-based SD gRNAs (Fig. 2), the toehold and the separator domain have a largely arbitrary sequence and can thus be used to achieve orthogonality. However, designing distinct trigger sequences based solely on different toehold domains may not be able to produce sufficiently orthogonal activation since toehold-free strand invasion into the switch domain of the SD gRNA can still occur due to duplex fraying[1]. Furthermore, with a toehold length of only 9 nt the sequence design space is limited.

We therefore created a second type of SD gRNAs (termed target-based SD gRNAs), in which we extended the switch

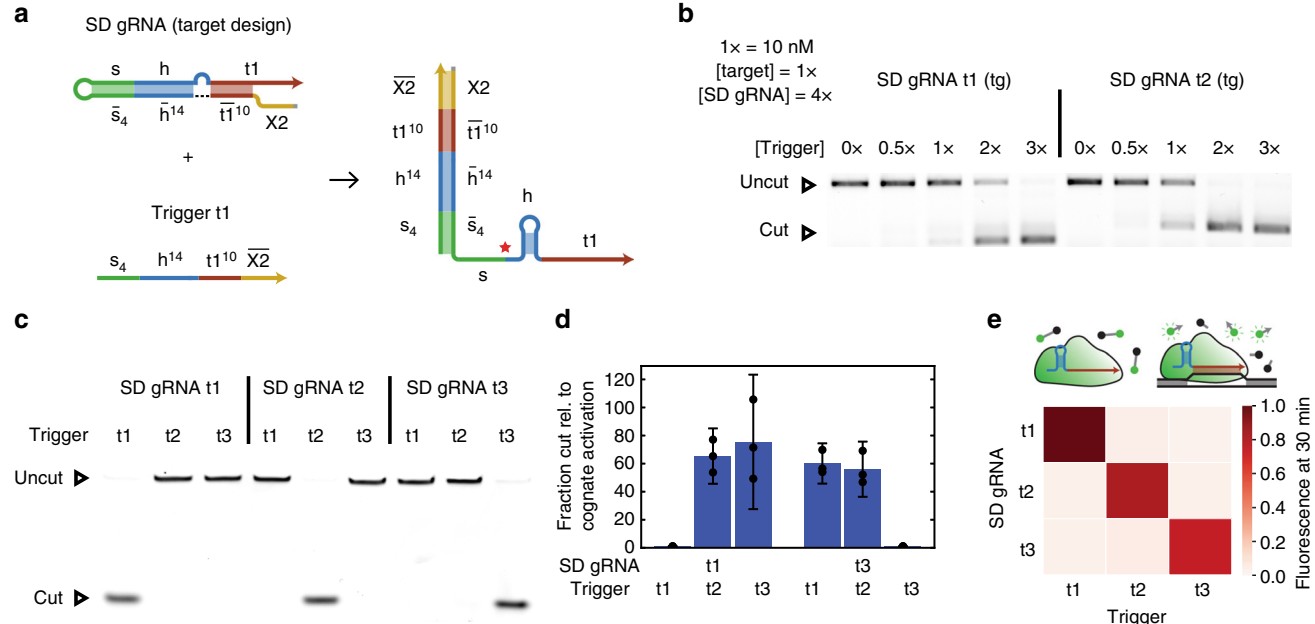

**Fig. 3** Target-based strand displacement gRNAs. **a** Design of target-based (tg) strand displacement gRNAs, for which part of the target domain is occluded. Five nt of the handle remain unpaired to avoid secondary structure in the trigger. **b** Agarose gel showing activation of target cutting by varying amounts of trigger for target-based strand displacement gRNAs (uncut: 1190 bp, cut t1: 357 bp, cut t2: 506 bp). **c** Denaturing PAGE showing orthogonal cutting of target dsDNA by SD gRNAs with three different target domains. **d** Fraction of cut target due to SD gRNA activation by the cognate trigger divided by the fraction of cut target due to the shown triggers as determined by gels as shown in (**c**). The quantification is based on the analysis of a denaturing PAGE (cf. Fig. S5f). For example, the second bar from the left shows that trigger t2 leads to roughly 1/60 as much cutting as trigger t1 for SD gRNA t1. Non-cognate activation for target sequence t2 was too small to be quantified (uncut: 79 nt, cut: 22 nt) ($N = 3$, $t$-distribution two-sided 90% confidence interval). **e** ssDNase assay after 30 min of reaction time for combinations of SD gRNAs with different triggers ($N = 3$). Source data are provided as a Source Data file

domain to also include 10 nt of the target sequence (Fig. 3a), effectively introducing an insulating domain that stops toehold-free strand displacement. For our initial design of orthogonal SD gRNAs, we deliberately left the separator domain fixed to demonstrate that different target domains already provide sufficient orthogonality. When designing circuits with artificial Cas12a targets, the sequence of the target is practically arbitrary[39]. When targeting a naturally occurring sequence, the target sequence obviously must be chosen from the fixed set of potential target sites on that gene. Since this sequence will be different for different SD gRNAs, however, this also does not interfere with the construction of sufficiently orthogonal SD gRNAs.

Initial design attempts wherein the switch only paired a small part of the handle and the target sequence showed substantial leaky cutting and ssDNase activity already in the absence of trigger RNA (Supplementary Fig. 5). The probable reason for this behavior is the transient formation of the handle hairpin, which occurs when the regions adjacent to the gRNA processing site are not strongly paired. In our final design of target-based SD gRNAs, we therefore occluded both the handle and target domain and put the toehold adjacent to the target-complementary region of the switch domain. Accordingly, an activating trigger RNA must displace this domain first before it can invade the handle domain which is partially shared between different gRNAs.

In total, three factors contribute to the orthogonality in activation of target-based SD gRNAs: First, an orthogonal toehold slows the kinetics of non-cognate trigger RNA strand displacement. Second, different target domains stop non-cognate triggers from accessing the handle domain. Finally, orthogonality is introduced even for nominally fixed sequences by varying the usage of GU wobble base pairs (see Supplementary Note 3 and Supplementary Fig. 13 for a detailed explanation). Sequence design was greatly facilitated by the constrained multistate

sequence design and test tube functionality of NUPACK, which allows for explicit specification of orthogonality between SD gRNAs and non-cognate triggers[11].

As shown in Fig. 3b and Supplementary Fig. 5e, target-based SD gRNAs generate roughly half as much cut target for the same amount of trigger as the handle-based designs. As desired, different SD gRNAs designed using the method described above are orthogonal in their activation by their respective triggers (Fig. 3c). Analysis of the cutting assay indicates that there is at least a 50-fold difference between the activity of target-based SD gRNAs when triggered by their cognate triggers compared with undesired activity due to inherent leak or in the presence of non-cognate triggers (Fig. 3d, Supplementary Fig. 5f).

The indiscriminate ssDNase activity of Cas12a allows for a particularly sensitive readout for SD gRNA activity as a single activated Cas12a-gRNA complex can catalyze the cleavage of thousands of target ssDNAs[23]. Here, too, the SD gRNAs show orthogonal activation (Fig. 3e). At the concentrations used for our experiments, full digestion of the target ssDNA occurs only a few minutes more slowly than for a regular gRNA (Supplementary Fig. 6a). Notably, the combination of trigger t2 and SD gRNA t3 shows about twice as much leaky activation than most of the other combinations (Supplementary Fig. 6b). In fact, the SD gRNA for this target sequence already showed substantial leak in an earlier target-based SD gRNA design (Supplementary Fig. 5d), indicating that certain target sequences are more problematic than others. We hypothesize that leaky activation in this case is caused by the transient formation of the handle structure. In fact, NUPACK predicts that target sequence t3 has substantially stronger interactions with the separator- and handle-complementary parts of the switch domain of its SD gRNA than the target sequences t1 and t2 have for their corresponding SD gRNAs.

**Implementing multi-input logic SD gRNAs.** As previously shown for toehold switch riboregulators, splitting an RNA trigger molecule into multiple parts facilitates the rational construction of trigger-based AND gates[19]. We therefore investigated whether a similar approach could also be applied to SD gRNAs and thus enable logical control of Cas12a activity (Fig. 4). We first chose to

implement an AND gate for handle-based gRNAs with target domain t1. To this end the triggers are split in two and are extended by two complementary AND domains which can hybridize with each other but not with either the gRNA toehold or the rest of the trigger (Fig. 4a). When both parts p1 and p2 of the AND trigger are present, the combined trigger RNA shows

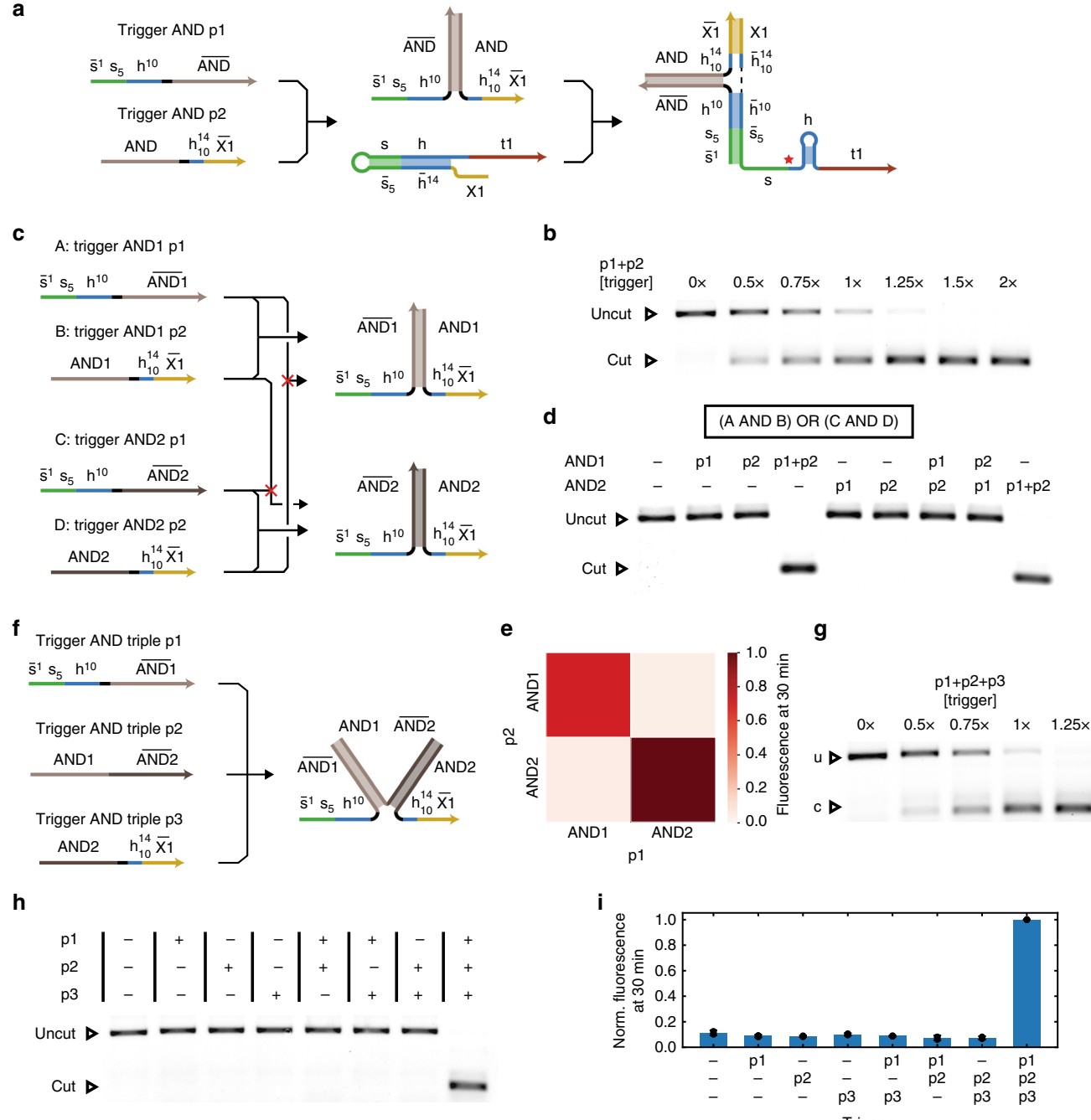

**Fig. 4** Trigger AND gates for SD gRNA activation. **a** Principle of AND gates for SD gRNAs. The trigger is split in two and can reassemble based on a hybridization domain. The reassembled trigger activates the SD gRNA. **b** Agarose gel assaying target cutting for different amounts of the two assembled trigger parts (uncut: 1190 bp, cut: 357 bp). **c** Orthogonal hybridization domains lead to orthogonal AND gates. **d** Agarose gel assaying target cutting for different combinations of trigger RNAs. Only two cognate trigger parts lead to significant cutting activity (uncut: 1190 bp, cut: 357 bp). **e** ssDNase assay for orthogonal AND triggers ($N = 3$). **f** The principle of the two-input AND gate can be extended to a three-input AND gate by distributing the hybridization domains. **g** Agarose gel assaying target cutting for different amounts of the three assembled trigger parts (uncut: 1190 bp, cut: 357 bp). **h** Agarose gel demonstrating functionality of the three-input AND gate (u, uncut: 1190 bp, c, cut: 357 bp). **i** ssDNase assay of gRNA activation by different combinations of triple AND triggers. Only all three triggers together lead to an activation significantly above the background level ($N = 3$, t-distribution two-sided 90% confidence interval). Source data are provided as a Source Data file

similar transfer characteristics for target cutting as the full-length trigger, though the kinetics are slightly slower (Fig. 4b, Supplementary Fig. 6c). The fluorescence data demonstrates that the time scale of activation and Cas12a binding of SD gRNAs is on the order of minutes and should not limit applications.

To show that the hybridization domain is indeed necessary for the activity of the assembled trigger, we designed an additional set of AND triggers with the same split switch and toehold domains, but with orthogonal AND domains, resulting in two orthogonal AND gates (Fig. 4c). Even in the context of the sensitive ssDNase assay, addition of a single trigger part of either AND gate does not lead to a significant activation of the SD gRNA (Supplementary Fig. 6c, d). In this sense, the AND gates show almost ideal digital behavior. Importantly, this still holds when mixing parts of orthogonal AND gates (Fig. 4e). Adding parts p1 and p2 of orthogonal AND gates is equivalent to the absence of any trigger. Taken together, the two orthogonal AND gates constitute the logic function (A AND B) OR (C AND D) (Fig. 4d).

The split-input concept can be extended to a three-input trigger AND gate (Fig. 4f). Here, too, cutting is only observed for the addition of all three parts of the trigger (Fig. 4h), and the ssDNase assay does not show significant activation for any combination of two trigger parts, especially for parts p1 and p3, which contain the entire sequence of the original trigger (Fig. 4i). In this way, a disjunctive normal form can be readily translated into trigger RNAs for SD gRNAs. We did not implement NOT as an explicit RNA strand. As demonstrated by Green et al., this could likely be accomplished with an RNA that is fully complementary to a trigger RNA[19].

**Natural RNA sequences as input triggers**. As our SD gRNAs contain a fixed sequence domain (namely the handle domain), natural RNA molecules such as mRNAs—with a practically arbitrary sequence—cannot be used as trigger molecules directly. However, it is possible to adapt the design of the AND gate triggers to facilitate sensing of natural RNA sequences. This is achieved simply by placing the fixed sequences into the first (p1) input of the AND trigger, while treating the natural RNA sequence as the second (p2) input (Supplementary Fig. 7a). As an example for this scheme, we created an AND gate sensor for an mCerulean mRNA. When using a short subsequence of the natural mRNA molecule as the input, the activation kinetics of the SD gRNA was found to be as fast as for the engineered inputs, demonstrating that natural RNA sequences can indeed be sensed by split-input SD gRNAs. In contrast, kinetics is considerably slowed down when using the corresponding full-length mRNA molecule as the input (Supplementary Fig. 7b). To test whether the activation is specific for mCerulean mRNA, we used mCherry mRNA as a control input. Surprisingly, we found that the addition of non-cognate mRNA not only did not activate the SD gRNA, but even strongly reduced the leak reaction in the presence of SD gRNA and trigger part p1, indicating that secondary structure obscuring the sensed part of the mRNA probably is not the dominant effect responsible for the difference in kinetics for partial and full-length mRNAs.

**Implementing SD gRNAs for transcriptional regulation**. We next explored whether the SD gRNA concept developed in the previous paragraphs could also be applied for in vivo gene regulation in bacteria. To this end, we utilized the DNase inactive version of Cas12a (dCas12a), which had previously been shown to efficiently repress transcription when bound to the target strand of a gene[40].

In order to demonstrate transcriptional control via SD gRNAs in E. coli, we created a low copy number plasmid (≈15 copies)

containing gene sequences coding for the fluorescent protein mVenus and SD gRNAs targeting mVenus expression, each put under the control of a constitutive promoter. In addition, the plasmid coded for dCas12a, which was put under the control of an IPTG-inducible $P_{lacO1}$ promoter. The sequence coding for the trigger RNA was placed under a constitutive promoter onto a separate plasmid with a similar copy number. To test the effect of the absence of a cognate trigger, we used a plasmid containing a control trigger that was designed not to interact with the SD gRNAs. As mVenus expression is repressed in the presence of both SD gRNA and trigger RNA in this setting, each SD gRNA-trigger pair can be interpreted as a NAND gate for gene expression.

We initially tested an SD gRNA for target t1 (version A), which had exactly the same design as the one used for the in vitro experiments shown in Fig. 3. As demonstrated in Fig. 1, adding a hairpin to the 3′ end of a gRNA strongly decreases its activity. The SD gRNA is therefore transcribed with an additional AsCas12a handle at its 3′ end which is removed along with the terminator by dCas12a's RNase activity[25]. In the absence of trigger RNA, the design showed only a slight (leak) repression on the order of 20%, as judged from the mVenus expression level measured 4 h after induction. However, repression by the triggered SD gRNA was significantly less efficient than for the regular gRNA we used as a control, resulting in an on/off ratio of ≈10 (Fig. 5a, Supplementary Fig. 8, and Supplementary Note 5).

We surmised that under in vivo conditions the trigger did not displace the switch domain effectively enough, which could either be due to inefficient hybridization between trigger and toehold or due to slow strand displacement kinetics. We therefore designed two slightly altered versions of the SD gRNA target design (version B and version C). To address possible inefficient hybridization, both designs were equipped with a 12 nt toehold instead of the 9 nt toehold of version A.

To address the problem of possibly slow strand displacement kinetics, we introduced one and two kinetic insulators to versions B and C, respectively. These kinetic insulators consist of two unpaired nucleotides (internal mismatches) which can be bound by the trigger during displacement, but are not or only weakly bound within the switch domain. This splits up the switch domains into several parts that can be displaced separately while largely eliminating the possibility of reverse branch migration. Finally, the separator domain was released from sequence constraints and shortened to reduce the total length of the displaced duplex. Additional minor adjustments are explained in the Methods, Supplementary Note 6, and Supplementary Fig. 14.

Both version B and C SD gRNAs show no obvious leaky repression (as judged from mVenus expression 4 h after induction), and, more importantly, a strongly increased repression in the presence of trigger RNA (Fig. 5a, Supplementary Fig. 8). For both versions, repression is as effective as for a regular gRNA ('reg'). Accordingly, the dynamic range, as measured by the ratio of repression efficiency between induced samples containing SD gRNAs in the presence and absence of trigger, is as high as ≈100 4 h after induction.

Since version C displayed a performance slightly superior to version B (cf. Fig. 5a), we continued with version C SD gRNAs for further experiments. We chose three additional target sequences on the mVenus gene that were selected with the Cas12a target prediction tool DeepCpf1 (Fig. 5b)[41]. When activated by their cognate triggers, also these SD gRNAs show repression rates equivalent to those obtained with regular gRNAs, demonstrating that our use of dCas12a's RNA processing activity has the intended effect also in vivo. The intrinsic leak is close to zero for target sequences t1, t4, and t5, but noticeable for target sequence t6, which also has the lowest repression efficiency. The separator

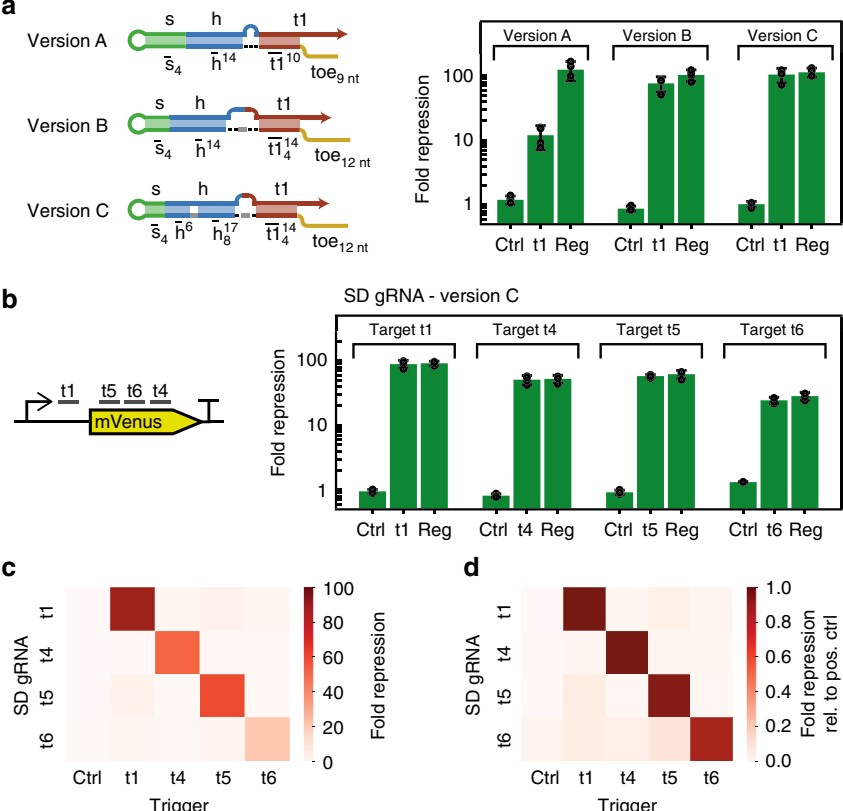

**Fig. 5** Transcriptional regulation via SD gRNAs and dCas12a in *E. coli*. **a** Repression of mVenus fluorescence by different versions of target-based SD gRNAs in *E. coli* with and without the expression of a cognate trigger. The design of the different versions is shown on the left. Version A is identical to the target-based SD gRNAs shown in Fig. 3. Expression of the dsDNase inactive mutant dCas12a is induced with IPTG and the fluorescence per OD at 4 h after induction is compared with the fluorescence per OD of uninduced cells containing the same SD gRNA plasmid and no cognate trigger. A regular dCas12a gRNA is used as a positive control for repression (ctrl: control trigger, t1: cognate trigger, reg: regular gRNA) ($N = 4$, $t$-distribution two-sided 90% confidence interval). **b** Fold repression by the SD gRNA version C design for four different target sequences on the mVenus gene at 4 h after induction (ctrl: control trigger, t1-t6: cognate triggers, reg: regular gRNA) ($N = 4$, $t$-distribution two-sided 90% confidence interval). **c** Crosstalk between different SD gRNAs and their triggers at 4 h after induction ($N = 4$). **d** Crosstalk between different SD gRNAs and their triggers at 4 h after induction normalized by the fold repression of the respective positive control gRNA. ($N = 4$) Source data are provided as a Source Data file

domain for this specific SD gRNA consists exclusively of AU and GU pairs, which may explain its relatively high leak.

While the function of the SD gRNA NAND gates is close to ideal when operated separately, their orthogonality depends strongly on the individual target sequence and SD gRNA design (Fig. 5c, Supplementary Fig. 9). For example, SD gRNA for target t4 has low crosstalk with all other triggers (<15% variation in repression level between induced and uninduced samples for the non-cognate triggers as compared with a >50-fold repression for the cognate trigger). On the other hand, SD gRNA t6 has significant crosstalk with the other triggers, consistent with its relatively high intrinsic leak. With a relative spurious repression level of SD gRNA t1 by the trigger for SD gRNA t5 of about 6%, SD gRNAs t1 and t5 display an appreciable level of crosstalk (Fig. 5d).

Similarity between target sequences could be one possible cause of crosstalk. Another possibility is cotranscriptional binding of the trigger to the switch domain. While in the in vitro experiments described above, the handle domain is always insulated from trigger binding by the target-complementary domain of the switch, in the in vivo experiments this is only true after the entire SD gRNA has been transcribed and folded. During transcription of the SD gRNA, trigger RNA could bind to the nascent handle-complementary domain of the switch, leading to a transient gRNA handle formation, dCas12a binding, and therefore leak.

**Multi-input logic in *E. coli*.** We next asked whether multi-input SD gRNAs can be implemented in *E. coli* according to the same principles as shown in vitro. We therefore constructed a trigger AND gate by placing the entire separator- and handle-cognate sequences on trigger part p1 and the toehold- and target-cognate sequences on trigger part p2 (Fig. 6a). As the relatively high spurious repression observed for some of the non-cognate triggers discussed above might also pose a problem for proper AND gate operation, the first trigger part was deliberately engineered to have some secondary structure to discourage potential cotranscriptional binding of the SD gRNA. This secondary structure is removed by the AND domain of the second trigger part upon binding and is expected not to slow down strand displacement kinetics.

As shown in Fig. 6b, as desired the individual trigger parts do not activate repression, which demonstrates that even fully complementary separator- and handle-cognate domains do not necessarily lead to leaky repression. When both trigger parts are present simultaneously, the SD gRNA is activated. With a 10-fold repression after 4 h, and 20-fold repression after 6 h, the repression level achieved by this particular design is weaker, however, than for a single trigger. This might be expected as the lifetime of RNAs in *E. coli* is relatively short (on the order of a few minutes), and activation of the AND gate requires hybridization of the two trigger parts. In consequence, the

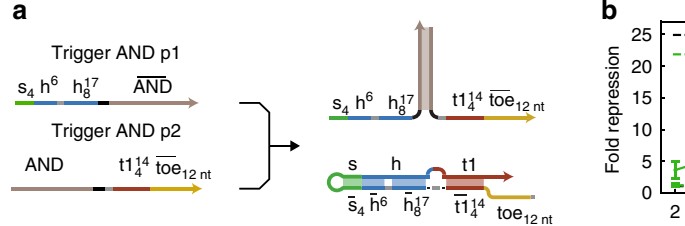
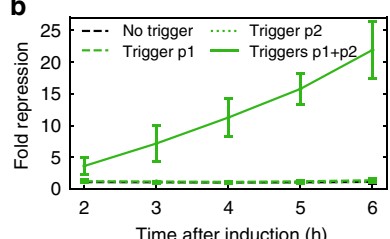

**Fig. 6** Multi-input SD gRNAs in *E. coli*. **a** Design of a trigger AND gate for the activation of SD gRNAs (version C). **b** Repression of mVenus at different time points after the induction of dCas12a for the trigger AND gate (black dashed line: no trigger, green dashed line: trigger p1, green dotted line: trigger p2, solid green line: triggers p1 + p2), ($N = 4$, $t$-distribution two-sided 90% confidence interval). Source data are provided as a Source Data file

effective concentration of fully assembled triggers and hence activated SD gRNAs may be considerably lower than when using a single trigger[42].

## Discussion

We established an approach to extend AsCas12a gRNAs with switchable domains which are controllable via RNA strand displacement. For this purpose, we first tested strategies for extending Cas12a gRNAs with additional sequence elements. While extension at the gRNA handle stem and the 3′ end abolished Cas12a activity, extension at the 5′ end leads to gRNAs with activities comparable to those of regular gRNAs, which is facilitated by the intrinsic RNA-processing activity of Cas12a. This extension method should prove generally useful for the alteration of gRNAs of members of the Cas12a family and might also be applicable to the gRNAs of Cas13a, which also performs gRNA processing.

Extending gRNAs with a switch domain that occludes the handle domain results in switchable gRNAs, which can be activated with appropriate trigger RNAs via toehold-mediated strand displacement. By itself, the handle-based design could already be used, e.g., for enzymatic amplification of the output of an in vitro sensing system which is based on strand displacement reactions[9]. By extending the occluding domain to sequester also parts of the target domain, multiple orthogonal triggers were constructed for gRNAs with different targets. These orthogonal SD gRNAs show essentially digital behavior—in the absence of trigger or the presence of an orthogonal trigger, they display only minimal leak, while in the presence of a sufficient amount of cognate trigger they fully cut their target DNA.

Splitting the trigger into multiple parts allows for the implementation of multi-input SD gRNAs. Here, we demonstrated two-input and three-input SD gRNAs, but a larger number of inputs is easily conceivable[19]. By using alternative and orthogonal interaction domains in the AND trigger parts, also OR logic could be realized, which implies that, in principle, arbitrary logic functions could be implemented with SD gRNAs.

Other strand displacement mechanisms have been previously employed for the detection of mRNA in vitro and in vivo[1,17,18]. Here, we have demonstrated that an in vitro two-input AND gate based on our SD gRNAs can also be used to sense natural RNA sequences. In the current design, the kinetics are still relatively slow for a full-length mRNA input. Further improving the sensing capabilities of SD gRNAs appears possible, however, and is an exciting direction for future work. For instance, using a three-input AND gate could potentially overcome slow kinetics and the remaining sequence constraints by using mRNA as an assembly scaffold for the trigger parts. For sensing low abundance mRNAs, this could be combined with a scheme based on catalytic hairpin assembly[43].

As shown in this work, SD gRNAs can also be adapted for transcriptional regulation in *E. coli* using dCas12a. The use of

dCas12a's gRNA processing capability renders activated SD gRNAs as effective as regular gRNAs, leading to a dynamic range of roughly 50–100 for several of the target sequences tested—a number that could probably be increased further for gRNAs with optimized targets.

Further improvement of the orthogonality of in vivo SD gRNAs could be achieved by pairing a larger fraction of the target domain, via more stringent sequence design, or by introduction of secondary structure into triggers to discourage cotranscriptional binding. For multi-input SD gRNAs the leak is very small and the primary focus for future research should be laid on the improvement of repression efficiency, which could be critical for sensing applications. We surmise that the RNase activity of dCas12a should be especially useful when detecting mRNAs in vivo—without gRNA processing the sensed mRNA would remain attached to the gRNA, probably reducing repression efficiency considerably.

This work established SD gRNAs for in vitro strand displacement circuits and for transcriptional regulation in *E. coli*. Since dCas12a fusion proteins have already been shown to function as transcriptional activators or repressors in mammalian cells, the general principles explored in this article could also be applicable in eukaryotic cells[44].

## Methods

**Design of gRNAs and triggers**. The design of the stem extension, 3′ extension and 5′ extension gRNAs was performed by hand and verified by NUPACK. The design of all strand displacement gRNAs and triggers was performed using NUPACK[11]. Design considerations are outlined in the Supplementary Notes. NUPACK design files are supplied separately as Supplementary Software. The sequences for all components used can be found in Supplementary Data 1 to 4 and Supplementary Table 1.

**Synthesis of DNA templates and targets**. DNA template sequences were constructed by adding a T7 promoter in front of the designed RNA sequences. The resulting template sequences were split into two parts with sufficient overlap. The two parts were ordered as single-stranded DNA oligos from Eurofins Genomics. The double-stranded template was produced by annealing the strands and filling in the single-stranded regions with Phusion HF PCR Master Mix (New England Biolabs) by cycling between the melting temperature of the strands and 72 °C twenty times. The resulting DNA templates were column purified using a Monarch PCR&DNA Cleanup Kit (New England Biolabs) and quantified by their absorption on a NanoPhotometer (Implen). The length of the templates was verified in a native TBE 12% polyacrylamide (29:1 acrylamide/bis-acrylamide) gel run at 120 V for 45 min by comparing with a dsDNA ladder (Low Molecular Weight DNA Ladder, New England Biolabs). Short single target dsDNAs were ordered as overlapping ssDNA oligos and prepared as described above. Long dsDNAs containing multiple targets were PCR amplified from a plasmid. To attach the fluorescent label (Atto550), the targets were PCR amplified with a primer containing the label at the 5′ end.

**Transcription and purification of gRNAs and RNA triggers**. The transcription mixture contained ~80 nM of DNA template, 1× T7 RNA Polymerase buffer (40 mM Tris-HCl, 6 mM MgCl$_2$, 1 mM DTT, 2 mM spermidine, pH 7.9), 4 mM of each rNTP, 12 mM MgCl$_2$, and 1 U/µl T7 Polymerase (all New England Biolabs). The transcription mix was incubated at 37 °C for 4 h. For very impure transcriptions (cf. Supplementary Fig. 4), RNA was gel purified. The RNA was run on an 8 M urea

denaturing TBE 12% polyacrylamide (29:1 acrylamide/bis-acrylamide) gel at 120 V for 45 min, stained with SYBR Green II (ThermoFisher), and excised on a UV table. The gel slice was then purified using a ZR small-RNA PAGE Recovery kit (Zymo Research) according to the manufacturer's instructions. For RNAs with no significant bands of incorrect length, the RNA was phenol-chloroform purified using Phase Lock Gel Heavy (VWR) tubes. One volume of Roti-Aqua-P/C/I (Carl Roth) was mixed with the transcription reaction and the sample was centrifuged at $16{,}000 \times g$ for 5 min. One volume of chloroform (Carl Roth) was added, the sample was mixed, and centrifuged at $16{,}000 \times g$ for 5 min. The supernatant was transferred to a new tube, 0.1 volume of 3 M sodium acetate and three volumes of 99.8% ethanol were added. The mixture was kept at $-80\,^{\circ}\mathrm{C}$ for 1 h and centrifuged at $4\,^{\circ}\mathrm{C}$ and $16{,}000 \times g$ for 15 min. The supernatant was removed, 500 μl of 70% ethanol were added, and the sample was centrifuged at $4\,^{\circ}\mathrm{C}$ and $16{,}000 \times g$ for 5 min. The supernatant was removed, and the remaining liquid was evaporated. The pellet was resuspended in nuclease-free water. For both purification methods, the purified RNA was quantified by running it on an 8 M urea denaturing TBE 12% polyacrylamide (29:1 acrylamide/bis-acrylamide) gel at 120 V for 45 min and compared with an RNA ladder of known concentration (RiboRuler Low Range RNA Ladder, ThermoFisher) to determine its concentration and with a DNA ladder (Low Molecular Weight DNA Ladder, New England Biolabs) to determine its length.

**Cutting assays for gRNAs.** AsCas12a (containing nuclear localization sequences and C-terminal His tags) was ordered from IDT with a concentration of around 64 μM. Prior to use, the protein was diluted in 1× PBS to 5 μM, kept at $4\,^{\circ}\mathrm{C}$ and used within 24 h. The cutting reactions were performed by incubating 10 nM of atto550-labeled target with 150 nM of AsCas12a in 1× NEBuffer 3.1 (100 mM NaCl, 50 mM Tris-HCl, 10 mM $MgCl_2$, 100 μg/ml BSA, pH 7.9, New England Biolabs). The gRNA and trigger concentrations varied according to the experiment. For testing the extension strategies, we used ~100 nM of gRNA to be able to detect residual cutting for poorly working gRNAs. The trigger titration experiments used 40 nM of SD gRNA. The experiments assaying orthogonal cutting by target-based SD gRNAs or AND gate triggers used 40 nM of SD gRNA and 40 nM of trigger RNA. The experiment assaying induction of SD gRNA processing by the trigger RNA used 125 nM of gRNA and 375 nM of trigger.

The reaction mix was incubated at $37\,^{\circ}\mathrm{C}$ for 1 h. The mix was then kept at $95\,^{\circ}\mathrm{C}$ for 8 min and at $50\,^{\circ}\mathrm{C}$ for 5 min to detach Cas12a from the target. For reactions with long targets, the reaction mix was run on a 1.5% agarose gel at 120 V for 25 min. For reactions with short targets, the reaction mix was run an 8 M urea denaturing TBE 12% polyacrylamide (19:1 acrylamide/bis-acrylamide) gel at 120 V for 45 min. The gels were imaged in a Typhoon FLA 9500 (GE Healthcare Life Sciences). Uncropped gels are provided in the Source Data file.

To quantify the fraction of target that was cut, the area under the curve for the uncut band was determined using ImageJ. The area of the uncut and cut bands was used as the reference to determine the fraction of cut target.

**Fluorescence measurements for ssDNase assays.** For ssDNase assays, we used a short reporter DNA oligo with the sequence FAM-TTATT-BHQ1[23]. A single reaction had a total volume of 15 μl and was run with 1× NEBuffer 3.1, 7 nM of each target, 100 nM reporter oligo, 60 nM of AsCas12a, 7–12 nM of SD gRNA, and 30–60 nM of trigger concentration. Within one set of experiments, SD gRNA and trigger concentrations were the same for all SD gRNAs and triggers. The components were assembled at $4\,^{\circ}\mathrm{C}$, transferred into white tubes with clear caps (ThermoFisher) and measured at $37\,^{\circ}\mathrm{C}$ in a BioRad iCycler for 1 h.

The data were treated as follows: The baseline was subtracted by shifting the lowest fluorescence value for all samples to zero. The fluorescence was normalized by dividing all fluorescence curves by the highest value occurring in this experiment. For bar graphs, the fluorescence value 30 min after the start of the experiment was determined and used as the measured value for that bar graph.

**Bacterial strains and plasmid construction.** The full sequences of transcribed RNAs as well as predicted processed RNA sequences can be found in Supplementary Data 3. The plasmids used can be found in Supplementary Data 4.

Plasmids were constructed using a combination of standard cloning techniques. The gene for dAsCas12a (mutation D908A) was taken from a plasmid obtained as a kind gift from Ilya Finkelstein's lab[33]. The genes for all other inserts were ordered from IDT as gBlocks. We used a pSB3C5 vector to construct our SD gRNA plasmids. An SD gRNA plasmid contains a constitutively expressed *lacI* gene, dCas12a under the control of a pLacO1 promoter, mVenus with a constitutive J23104 promoter, and an SD gRNA with a constitutive J23110 promoter (J23102 for version A) and an L3S3P21 terminator[45]. Trigger plasmids were constructed from a pET21b vector. Single triggers are expressed from a pTetO1 promoter with an L3S1P56 terminator. The strain we use does not contain a Tet repressor, making the pTetO1 promoter constitutive in this case. For multi-input triggers, trigger part p1 has a J23118 promoter and a L3S1P56 terminator and trigger part p2 has a J23108 promoter and a L3S2P21 terminator.

We used the DH5α strain to assemble and purify the plasmids, and *E. coli* from strain MG1655 for testing our circuits. MG1655 was chosen to be close to wild-type bacteria. T7 RNA polymerase was not used to reduce the metabolic load of the circuits on the bacteria.

**Transcriptional repression by SD gRNAs.** MG1655 cells were transformed with a pSB3C5 SD gRNA plasmid and a pET21b trigger plasmid using heat shock. A single colony was picked and grown for 12–14 h in LB medium supplemented with 100 μg/ml carbenicillin and 25 μg/ml chloramphenicol. The cultures were diluted 1:100 in M9 medium (6.78 g/L $Na_2HPO_4$, 3 g/L $KH_2PO_4$, 1 g/L $NH_4Cl$, 0.5 g/L NaCl, 2 mM $MgSO_4$, 100 μM $CaCl_2$, 20 mM glucose, 0.2 wt% casein hydrolysate, 0.2 g/l thiamine) supplemented with 50 μg/ml carbenicillin and 12.5 μg/ml chloramphenicol and grown for 3–4 h to an OD of ~0.5. The resulting cultures were diluted to an OD of 0.035 in the same M9 medium and divided into subcultures which were either left uninduced or were induced with 60 μM of IPTG. The subcultures were divided into technical duplicates or triplicates into a 96-well plate, incubated at $37\,^{\circ}\mathrm{C}$ and shaken at 500 rpm in a FLUOstar Omega Microplate Reader. OD and fluorescence data were measured. The times shown refer to the beginning of the Microplate Reader measurement.

For data evaluation, technical replicates were averaged. The OD and fluorescence values of M9 medium were subtracted from those of the samples. To determine the fold repression of a given sample, the fluorescence per OD of the corresponding uninduced SD gRNA sample with a control trigger at a given time point was divided by the fluorescence per OD of that sample. For example, to determine the fold repression of the control gRNA and SD gRNA version A with and without cognate trigger, the fluorescence per OD value of the uninduced version A sample with a control trigger was divided by the induced fluorescence per OD of these samples. The dynamic range was determined by dividing the fold repression of the sample with the cognate trigger by the fold repression of the sample without cognate trigger.

**Statistical analysis.** When quantitative data are shown, they are typically derived from averaging three individual experiments. The exact number of replicates is given in the corresponding figure. Analysis of data was performed using Python. The error shown is the two-sided 90% confidence interval based on Student's *t*-distribution. Individual measurements are shown as black dots in the graphs.

**Reporting summary.** Further information on research design is available in the Nature Research Reporting Summary linked to this article.

## Data availability

All the conclusions drawn in this paper are derived from data shown either in the main text or in the Supplementary Information. The sequences for all used components can be found in Supplementary Data 1–4 and Supplementary Table 1. The source data underlying Figs. 2–6 and Supplementary Figs. 3 and 5–9 are provided as a Source Data file. It is also available from the corresponding author upon reasonable request.

## Code availability

NUPACK design files are supplied as a Supplementary Software file.

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

## Acknowledgements

We thank Ilya Finkelstein for kindly providing a plasmid containing dCas12a. We thank Enikö Baligács for assisting with preliminary measurements of the in vivo system and Daniela Ziegler for proofreading the manuscript. We gratefully acknowledge funding by the European Research Council (project AEDNA, grant number 694410) and by the BMBF through the EraSynbio network (project UNACS, grant no. 031L0011).

## Author contributions

L.O. and F.C.S. conceived the project. L.O. designed the sequences, performed the experiments and analyzed the data. F.C.S. supervised the project. L.O. and F.C.S. wrote the paper.

## Additional information

**Competing interests:** The authors declare no competing interests.

