## [Peer Review File · Nature Communications]

Reviewers' comments:

Reviewer #1 (Remarks to the Author):

In this article, the authors employed several unique features of Cas12a (i.e. the 5' pre-crRNA processing activity, target DNA cleaving activity and the collateral cleavage feature on non-pairing ssDNA substrates) and the RNA strand displacement method to successfully create logic gates, including both the two-input and the three-input AND gates. In general, this work introduces a new idea for regulation of the activities of Cas12a, which may include both the cleavage activity, the regulatory activity and the potential trans-cleavage activity, and would therefore be of interest to scientists in the fields of CRISPR, transcriptional regulation, etc. However, its in vivo application can be performed to more convincingly demonstrate the importance of the work.

Points:

- (1) Cpf1 is an old name, and therefore Cas12a should be used instead.
- (2) Both the sequences and their names cannot be recognized, and should be re-organized.
- (3) Some key refs are missing, including two articles describing the Cas12a trans-cleavage: S. Y. Li et al., CRISPR-Cas12a has both cis- and trans-cleavage activities on single-stranded DNA. *Cell Res* 28, 491-493 (2018). ; S. Y. Li et al., CRISPR-Cas12a-assisted nucleic acid detection. *Cell discovery* 4, 20 (2018).
- (4) The species name should be italicized, for example the "Francisella novicida" in line 109.
- (5) Based on the present results, I do agree that the designed AND gate systems work well in vitro. However, the in vivo system could be more complicated and there would be a large number of interference RNAs. Therefore, at least one example of in vivo circuits can be given for either genome editing or gene transcription or epigenetics study, with which the importance and the robustness of the work will be greatly improved.
- (6) Besides, the present in vitro systems were analyzed within 1 h, which is enough for in vitro assays. However, the in vivo circuits will not react as promptly as the in vitro assays, and it can be necessary to analyze the undesired activity (e.g. either due to the inherent leak or with the input of non-cognate triggers) for a longer time (e.g. in Figure S4e).

Reviewer #2 (Remarks to the Author):

In this study, the authors showed strand displacement-mediated activation of gRNA can regulate the DNA cleavage by Cpf1 by trigger-RNA-dependent manner. Activity of AsCpf1 gRNA can be repressed by attaching switch domain that abolishes secondary structure of gRNA. They succeeded in activating the repressed gRNA via RNA strand displacement by trigger RNA, and they termed this gRNA as SD gRNA. They further engineered SD gRNA to be activated by orthogonal multiple RNAs and built AND logic gate in vitro cell-free system. However, the strand displacement principles used in this article is very similar to the previously published methods (e.g. Green, A. A. et al. *Nature*, 2017). Also, authors did not show that SD gRNA could work in living cells, leaving broad gap between their results and the problem they wanted to address (in line 48 to 51). Overall, their experiments clearly showed that SD gRNA could work as they designed in vitro, but significance and novelty are limited as pointed above and the current manuscript is not suitable for publication in *Nature Communications*.

Major points:

1. In the introduction (in line 48 to 51), the authors said the advantage of this system against the previously reported strand displacement system is its regulatability of non-engineered parts (such as genomic mRNAs) and availability in eukaryotic cells. However, these two advantages have not been demonstrated yet. The authors should show the data to demonstrate these advantages, which would increase the impact of this study.
2. In Fig. 2d, the authors claim that "one molecule of trigger leads to the activation of

approximately one SD gRNA and the cutting of one target molecule." However, in Fig. 2c, they use the 125 nM of SD gRNA and 375 nM of trigger RNA, in which they observed the unprocessed SD gRNA. This suggests that one trigger RNA molecule cannot activate the same amount of SD gRNA. Also, they used 40 nM of SD gRNA and 10 nM of trigger RNA, greatly different condition from the Fig. 2c, which does not use 1 to 1 ratio of SD gRNA and trigger RNA. Therefore, there is no convincing experimental evidence to support author's claim. Besides, the possibility that one molecule of activated SD gRNA can cut multiple target DNA should also be examined.

3. In line 169, the authors say "we deliberately left the spacer domain fixed for our designs." However, the reason why they left the spacer domain fixed are not specified. Explaining the reason why they fixed or experimentally showing the programmability of this domain would be needed since they discuss "Randomizing the spacer would increase the sequence space" in line 274 to 275 and this is the potential merit of this device.

4. To claim the generality of the SD gRNA design, I think that seemingly sequence-dependent leak activity shown in Fig. S5b and Fig S4d should be investigated. Discussing the different characteristics of t3 sequence from t1 and t2 would help, but designing other sequences would be needed to understand the mechanism.

5. In Fig. S5c, while "AND2 p1 + AND2 p2" is shown, there is no "AND1 p1 + AND1 p2". As we can see in Fig. 4e, the activities of "AND2 p1 + AND2 p2" and "AND1 p1 + AND1 p2" are apparently different, which imply the difference of kinetics between these two conditions. Therefore, the author should also show the kinetics of "AND1 p1 + AND1 p2".

6. In Fig. S5c, the maximum fluorescence of "AND2 p1 + AND2 p2" is higher than that of the positive control (regular trigger), although the kinetics of the former is slower than the latter. Some explanations about this phenomenon are needed.

Minor points:

1. The term "spacer" they used to describe the sequence between handle and switch in SD gRNA was confusing since DNA-targeting sequence is also called spacer in CRISPR system and I recommend them to use other term. Also, I could not figure out what does "spacer" mean in the text "For single-stranded extensions, we tested a random, unstructured single-stranded sequence (ss v1), the full spacer of AsCpf1 (ss v2) and the full spacer taken from Francisella novicida Cpf1 (FnCpf1) (ss v3)." I recommend to use widely accepted terms and not to use the same term for different purposes in an article to help readers understand.

2. There is no description about the target of gRNAs used in Fig. 1. If t1 was the target of these gRNAs, the authors should describe it in page 4 or the legend of Fig. 1.

3. In line 190 to 191, they say "Finally, orthogonality is introduced even for fixed sequences by varying the usage of GU wobble base pairs," but it was not easy to immediately understand how they introduced orthogonality. Thus, it may be better to have figure showing the example.

4. Since they say "only slightly lower levels of activity," quantitative analyses on Fig. 3b would be needed.

5. It is hard to understand the sample sets of Fig. 3d.

6. In line 288 to 291, the authors "say Instead, building an mRNA sensor would either require the utilization of a toehold exchange reaction or the use of the sensed RNA as part of an AND gate, which perhaps is the most promising approach due to the lack of sequence constraints." I think they need references here that utilized a toehold exchange reaction or the sensed RNA as part of

an AND gate.

7. They should clearly discuss the obstacles to achieve a “universal nucleic acid computation framework capable of sensing and regulating the production of any RNA molecule in a cell” , since this article only showed in vitro results.

Point-by-point-response to the reviewer comments on

“Switching the activity of Cas12a using guide RNA strand displacement circuits”

We would like to thank the reviewers for providing their critical and insightful comments. In response, a considerable number of new and improved SD gRNAs for *in vivo* application in bacteria were designed, and a large number of corresponding experiments were conducted. We believe that this – as suggested by the reviewers – significantly improved the importance of our work.

Reviewers' comments:

Reviewer #1 (Remarks to the Author):

- In this article, the authors employed several unique features of Cas12a (i.e. the 5' pre-crRNA processing activity, target DNA cleaving activity and the collateral cleavage feature on non-pairing ssDNA substrates) and the RNA strand displacement method to successfully create logic gates, including both the two-input and the three-input AND gates. In general, this work introduces a new idea for regulation of the activities of Cas12a, which may include both the cleavage activity, the regulatory activity and the potential trans-cleavage activity, and would therefore be of interest to scientists in the fields of CRISPR, transcriptional regulation, etc. However, its *in vivo* application can be performed to more convincingly demonstrate the importance of the work.

Reply:

We thank the reviewer for this generally positive assessment.

We submitted the original version of the text with *in vitro* work only as we believe that this part of manuscript is, by itself, of great interest to readers in the field of molecular programming and nucleic acid computing, for example as an enzymatic amplification scheme for strand displacement circuits. Only very little of the work in molecular programming/strand displacement computing is performed in live cells.

Nevertheless, we of course agree that the demonstration of *in vivo* work would make our concept more convincing and, in fact, the design principles employed were in fact developed with the possibility of an *in vivo* application in mind.

In the revised version, we have therefore performed the requested *in vivo* work and would like thank the reviewer for suggesting this, since we believe it has greatly improved the quality of our manuscript. Specifically, we demonstrate the concept of SD gRNAs for application in CRISPR interference via dCas12a in *E. coli* and show that our SD gRNAs have repression efficiencies equivalent to wild-type gRNAs.

We demonstrate that several SD gRNAs can be constructed in an orthogonal manner, even though perfect orthogonality between all our *in vivo* SD gRNAs still poses a challenge. Our SD gRNA based AND gate also works *in vivo*. It shows a lower repression efficiency than single triggers, but still shows a very good performance overall.

As a comparison, we would like direct the reviewer to complementary work on dCas9 gRNAs which was released while this paper was in review (K.-H. Siu, W. Chen, Riboregulated toehold-gated gRNA for programmable CRISPR-Cas9 function. *Nature chemical biology* (2018).). The riboregulated gRNAs in this work do not have a shared domain, which makes it easier to employ orthogonality than our current design. However, probably due to dCas12a's RNA processing activity, our repression efficiency is an order of magnitude better. In fact, even the repression for our *in vivo* AND gate is similar to the best performing riboregulated gRNAs in the other work.

Points:

(1) Cpf1 is an old name, and therefore Cas12a should be used instead.

Reply: We have replaced all occurrences of 'Cpf1' by 'Cas12a'.

(2) Both the sequences and their names cannot be recognized, and should be re-organized.

Reply: We have split the Supplementary Tables into four parts, two for DNA and RNA sequences for the *in vitro* part of the manuscript, one for plasmids, and one for *in vivo* RNAs. We have checked that all sequence names in this table and the text match to make sure the relevant sequences can be easily identified.

(3) Some key refs are missing, including two articles describing the Cas12a trans-cleavage: S. Y. Li et al., CRISPR-Cas12a has both cis- and trans-cleavage activities on single-stranded DNA. *Cell Res* 28, 491-493 (2018). ; S. Y. Li et al., CRISPR-Cas12a-assisted nucleic acid detection. *Cell discovery* 4, 20 (2018).

Reply: We thank the reviewer for pointing out these references, which were added at appropriate positions in the text.

(4) The species name should be italicized, for example the "Francisella novicida" in line 109.

Reply: The names were italicized.

(5) Based on the present results, I do agree that the designed AND gate systems work well *in vitro*. However, the *in vivo* system could be more complicated and there would be a large number of interference RNAs. Therefore, at least one example of *in vivo* circuits can be given for either genome editing or gene transcription or epigenetics study, with which the importance and the robustness of the work will be greatly improved.

Reply: We have implemented our system in *E. coli* to demonstrate that it works *in vivo*. The reviewer's intuition about the AND gate was correct: The repression efficiency of the *in vivo* AND gate is indeed lower than that of single triggers. However, with a repression efficiency of ~ 10-20 fold the *in vivo* AND gate still works well overall.

(6) Besides, the present *in vitro* systems were analyzed within 1 h, which is enough for *in vitro* assays. However, the *in vivo* circuits will not react as promptly as the *in vitro* assays, and it can be necessary to analyze the undesired activity (e.g. either due to the inherent leak or with the input of non-cognate triggers) for a longer time (e.g. in Figure S4e).

Reply: In our *in vivo* system, the inherent leak remains low even for the longer periods required (the repression was analyzed 4 hours post induction). For non-cognate triggers, there is indeed significant leak for some SD gRNAs and little for others. As discussed in the text, we suspect that potential cotranscriptional binding of triggers to nascent SD gRNAs is responsible for the leak.

Reviewer #2 (Remarks to the Author):

In this study, the authors showed strand displacement-mediated activation of gRNA can regulate the DNA cleavage by Cpf1 by trigger-RNA-dependent manner. Activity of AsCpf1 gRNA can be repressed by attaching switch domain that abolishes secondary structure of gRNA. They succeeded in activating the repressed gRNA via RNA strand displacement by trigger RNA, and they termed this gRNA as SD gRNA. They further engineered SD gRNA to be activated by orthogonal multiple RNAs and built AND logic gate in vitro cell-free system. However, the strand displacement principles used in this article is very similar to the previously published methods (e.g. Green, A. A. et al. Nature, 2017).

Also, authors did not show that SD gRNA could work in living cells, leaving broad gap between their results and the problem they wanted to address (in line 48 to 51). Overall, their experiments clearly showed that SD gRNA could work as they designed in vitro, but significance and novelty are limited as pointed above and the current manuscript is not suitable for publication in Nature Communications.

Reply:

We thank the reviewer for this assessment, but we disagree, of course, that our work is not novel. Our work represents the first application of toehold-based strand displacement principles to CRISPR-associated proteins with an intrinsic pre-crRNA processing capability, such as Cas12a.

Strand displacement processes (on which one of the authors is working for almost 20 years) inevitably bear similarities as they all share the same basic principles. In this sense, our work is also similar to other, earlier work, of course, just as the toehold switches are.

In greater detail, however, the design of our SD gRNAs, specifically, has virtually no overlap with the toehold switches except for using strand displacement as a general concept. The only exception is design of the AND gate, which is indeed inspired by that by Green, et al., Nature, 2017, but which we acknowledge in the relevant paragraph.

Further, there are fundamental differences between toehold switches and our system. Toehold switches – placed in the 5' UTR of mRNAs - regulate gene expression at the translational level, whereas CRISPR-associated proteins such as Cas12a act as nucleases or can be used for transcriptional regulation (as now also shown in this work). This opens up the possibility to create “RNA in – RNA out” gene circuits, which is not possible with toehold switches (which always require a translation step). Furthermore, as translation is not required for readout, our system can be combined much more easily to in vitro strand displacement circuits.

Also, in vivo applications were not the primary “problem we wanted to address” (as our in vitro work should already be of considerable interest to the molecular programming community), but that was – at the stage of our first submission – meant as an outlook. However, we agree with this reviewer (and also reviewer #1) that our work would gain significance and the paper would be much more contentful if we showed in vivo applications. For this reason, we considerably expanded our manuscript and designed a series of new SD gRNAs for in vivo applications – specifically, we chose to demonstrate transcriptional regulation in E.coli with the help of the DNase-dead mutant dCas12a. For a general overview of our new results now included in the paper, we refer the reviewer to our introductory paragraph in the response to Reviewer #1.

Major points:

1. In the introduction (in line 48 to 51), the authors said the advantage of this system against the previously reported strand displacement system is its regulatability of non-engineered parts (such as genomic mRNAs) and availability in eukaryotic cells. However, these two advantages have not been demonstrated yet. The authors should show the data to demonstrate these advantages, which would increase the impact of this study.

Reply: These lines in the introduction were meant as an early outlook to possible future applications rather than the overall goal of the study. The goal of the study was to implement the control of Cas12a gRNAs for *in vitro* molecular computing. As mentioned above, however, we now demonstrate the applicability of our design for transcriptional regulation in *E. coli*. Implementing the concept in eukaryotic cells necessarily would have a much higher complexity and accordingly would take far more time.

The regulatability of non-engineered parts is now shown, in principle, by demonstrating the control of protein expression from a plasmid (we target “natural” sequences of fluorescent proteins). We did not implement the control of genomic mRNAs in this work. However, the control of unmodified genes on the genome by CRISPR proteins is very well established and to our mind does not represent a particularly controversial claim as an outlook.

2. In Fig. 2d, the authors claim that “one molecule of trigger leads to the activation of approximately one SD gRNA and the cutting of one target molecule.” However, in Fig. 2c, they use the 125 nM of SD gRNA and 375 nM of trigger RNA, in which they observed the unprocessed SD gRNA. This suggests that one trigger RNA molecule cannot activate the same amount of SD gRNA. Also, they used 40 nM of SD gRNA and 10 nM of trigger RNA, greatly different condition from the Fig. 2c, which does not use 1 to 1 ratio of SD gRNA and trigger RNA. Therefore, there is no convincing experimental evidence to support author’s claim. Besides, the possibility that one molecule of activated SD gRNA can cut multiple target DNA should also be examined.

Reply: We thank the reviewer for noticing this. In response, we have added a measurement to the SI (Fig. S4) which shows that the behavior for a 1 to 1 ratio of SD gRNA and trigger (10 nM SD gRNA and 10 nM trigger) is very similar to the behavior of the 4 to 1 ratio of SD gRNA and trigger shown in Fig. 2d. Interestingly, the cutting saturates at around 90% of cut target. As the reviewer correctly pointed out, there remains some unprocessed SD gRNA even for a large excess of trigger in the gels showing SD gRNA processing (Fig. 2c). We believe this indicates that some of our purified SD gRNA is nonfunctional, possibly due to damage caused by the purification, and therefore was not bound or processed by Cas12a. Regarding multiple cutting by Cas12a, we have added two references which show that Cas12a is effectively a single turnover enzyme.

3. In line 169, the authors say “we deliberately left the spacer domain fixed for our designs.” However, the reason why they left the spacer domain fixed are not specified. Explaining the reason why they fixed or experimentally showing the programmability of this domain would be needed since they discuss “Randomizing the spacer would increase the sequence space” in line 274 to 275 and this is the potential merit of this device.

Reply: We agree with the reviewer that we did not sufficiently explain our motivation for leaving this domain fixed. The intent was to show that the target domain is sufficient for orthogonality in this context, which we now clarify in the text. For our sensor and our *in vivo* designs, we allowed for an arbitrary separator domain which demonstrates the desired programmability.

4. To claim the generality of the SD gRNA design, I think that seemingly sequence-dependent leak activity shown in Fig. S5b and Fig S4d should be investigated. Discussing the different characteristics of t3 sequence from t1 and t2 would help, but designing other sequences would be needed to understand the mechanism.

Reply: We have added a short paragraph discussing differences between the target sequences. Our *in vivo* design now includes three new target sequences, which shows that the principle is valid for six different target sequences in total, which we would consider sufficient for a tentative claim to generality.

5. In Fig. S5c, while “AND2 p1 + AND2 p2” is shown, there is no “AND1 p1 + AND1 p2”. As we can see in Fig. 4e, the activities of “AND2 p1 + AND2 p2” and “AND1 p1 + AND1 p2” are apparently different, which imply the difference of kinetics between these two conditions. Therefore, the author should also show the kinetics of “AND1 p1 + AND1 p2”.

Reply: As per the reviewer’s request, the curve for AND1 p1 + AND1 p2 has been added.

6. In Fig. S5c, the maximum fluorescence of “AND2 p1 + AND2 p2” is higher than that of the positive control (regular trigger), although the kinetics of the former is slower than the latter. Some explanations about this phenomenon are needed.

Reply: We thank the reviewer for noticing this point. This behavior was the result of an outdated calibration for a few of the wells of our BioRad iCycler. We have checked the calibration and updated the data accordingly, which makes most of the difference in the final levels in this case disappear. The remaining difference is most likely due to pipetting error. We have adjusted the other fluorescence data accordingly and found that our overall data and conclusions are not affected in any way.

Minor points:

1. The term “spacer” they used to describe the sequence between handle and switch in SD gRNA was confusing since DNA-targeting sequence is also called spacer in CRISPR system and I recommend them to use other term. Also, I could not figure out what does “spacer” mean in the text “For single-stranded extensions, we tested a random, unstructured single-stranded sequence (ss v1), the full spacer of AsCpf1 (ss v2) and the full spacer taken from Francisella novicida Cpf1 (FnCpf1) (ss v3).” I recommend to use widely accepted terms and not to use the same term for different purposes in an article to help readers understand.

Reply: We thank the reviewer for this comment and fully agree that our chosen terminology was highly unfortunate and bound to lead to confusion. We have therefore renamed the “spacer domain” to “separator domain”. The passage highlighted by the reviewer was especially unclear and was rewritten to make our meaning more apparent.

2. There is no description about the target of gRNAs used in Fig. 1. If t1 was the target of these gRNAs, the authors should describe it in page 4 or the legend of Fig. 1.

Reply: The target sequence was indeed t1, which we added as a label to Figure 1.

3. In line 190 to 191, they say “Finally, orthogonality is introduced even for fixed sequences by varying the usage of GU wobble base pairs,” but it was not easy to immediately understand how they introduced orthogonality. Thus, it may be better to have figure showing the example.

Reply: The connection between GU pairs and orthogonality is explained in Part 2.3.3 of the Supplementary Information, to which we now explicitly direct the reader in the main text (we have added a table of contents to the SI to make it more accessible). Since this is a bit of a side point for which we do not have any direct data, and which requires a rather lengthy explanation to understand, we do not explore it further in the main text. The reference to GU pairs in Part 2.3.3 is mostly present in the SI to point out the potential of using GU pairs in RNA strand displacement as an interesting design feature for other researchers in our field who might find it interesting to explore.

4. Since they say “only slightly lower levels of activity,” quantitative analyses on Fig. 3b would be needed.

Reply: A quantitative analysis was added in Fig. S5e.

5. It is hard to understand the sample sets of Fig. 3d.

Reply: We have reformulated the caption for Fig. 3d to make the shown data clearer.

6. In line 288 to 291, the authors “say Instead, building an mRNA sensor would either require the utilization of a toehold exchange reaction or the use of the sensed RNA as part of an AND gate, which perhaps is the most promising approach due to the lack of sequence constraints.” I think they need references here that utilized a toehold exchange reaction or the sensed RNA as part of an AND gate.

Reply: To our knowledge, AND gates have not previously been used for nucleic acid sensing in the way we intend. We have therefore added a simple demonstration of the use of an AND gate for sensing mRNA in Fig. S7. This proof-of-principle demonstrates how the complication of a fixed handle domain for sensing can be overcome.

For the toehold exchange reactions, we have added a more specific reference to the use of catalytic hairpin assembly for *in vivo* nucleic acid sensing.

7. They should clearly discuss the obstacles to achieve a “universal nucleic acid computation framework capable of sensing and regulating the production of any RNA molecule in a cell” , since this article only showed *in vitro* results.

Reply: The origin of that statement was that our project was partly funded from a project named similarly ... We agree, however, that this statement was indeed unnecessarily broad in its claims and we therefore deleted it. Nevertheless, in the revised version of the manuscript, which has greatly benefitted from the reviewers' suggestion, *in vivo* results have been added, and the potential for RNA sensing has been demonstrated.

REVIEWERS' COMMENTS:

Reviewer #1 (Remarks to the Author):

The current manuscript by Oesinghaus et al. has been much improved, especially with the implementation of the in vivo experiments for transcriptional regulation in *E. coli*, which definitely highlights the importance of their findings. As the other points raised in the first round of reviewing have also been satisfactorily addressed, I think the manuscript is suitable for publishing in NC.

Reviewer #2 (Remarks to the Author):

In the revised manuscript, the authors have fully addressed my previous concerns. I believe that the current manuscript is recommended for publication.